# Head-To-Head Comparison of Biological Behavior of Biocompatible Polymers Poly(Ethylene Oxide), Poly(2-Ethyl-2-Oxazoline) and Poly[N-(2-Hydroxypropyl)Methacrylamide] as Coating Materials for Hydroxyapatite Nanoparticles in Animal Solid Tumor Model

**DOI:** 10.3390/nano10091690

**Published:** 2020-08-27

**Authors:** Zbynek Novy, Volodymyr Lobaz, Martin Vlk, Jan Kozempel, Petr Stepanek, Miroslav Popper, Jana Vrbkova, Marian Hajduch, Martin Hruby, Milos Petrik

**Affiliations:** 1Institute of Molecular and Translational Medicine, Faculty of Medicine and Dentistry, Palacky University Olomouc, Hnevotinska 5, 779 00 Olomouc, Czech Republic; zbynek.novy@upol.cz (Z.N.); miroslav.popper@upol.cz (M.P.); jana.vrbkova@upol.cz (J.V.); marian.hajduch@upol.cz (M.H.); 2Institute of Macromolecular Chemistry AS CR, Heyrovskeho namesti 1888/2, 162 06 Prague 6, Czech Republic; lobaz@imc.cas.cz (V.L.); stepan@imc.cas.cz (P.S.); 3Department of Nuclear Chemistry, Faculty of Nuclear Sciences and Physical Engineering, Czech Technical University, Brehova 7, 115 19 Prague 1, Czech Republic; martin.vlk@fjfi.cvut.cz (M.V.); jan.kozempel@fjfi.cvut.cz (J.K.)

**Keywords:** poly(ethylene oxide), poly(2-ethyl-2-oxazoline), poly[N-(2-hydroxypropyl)methacrylamide], hydroxyapatite, nanoparticles, solid tumor, animal model

## Abstract

Nanoparticles (NPs) represent an emerging platform for diagnosis and treatment of various diseases such as cancer, where they can take advantage of enhanced permeability and retention (EPR) effect for solid tumor accumulation. To improve their colloidal stability, prolong their blood circulation time and avoid premature entrapment into reticuloendothelial system, coating with hydrophilic biocompatible polymers is often essential. Most studies, however, employ just one type of coating polymer. The main purpose of this study is to head-to-head compare biological behavior of three leading polymers commonly used as “stealth” coating materials for biocompatibilization of NPs poly(ethylene oxide), poly(2-ethyl-2-oxazoline) and poly[*N*-(2-hydroxypropyl)methacrylamide] in an in vivo animal solid tumor model. We used radiolabeled biodegradable hydroxyapatite NPs as a model nanoparticle core within this study and we anchored the polymers to the NPs core by hydroxybisphosphonate end groups. The general suitability of polymers for coating of NPs intended for solid tumor accumulation is that poly(2-ethyl-2-oxazoline) and poly(ethylene oxide) gave comparably similar very good results, while poly[*N*-(2-hydroxypropyl)methacrylamide] was significantly worse. We did not observe a strong effect of molecular weight of the coating polymers on tumor and organ accumulation, blood circulation time, biodistribution and biodegradation of the NPs.

## 1. Introduction

Nanoparticles (NPs) are very frequently studied as potential diagnostics, therapeutics or theranostics for especially cancer applications (to date over 30,000 articles according to Web of Science). For diagnostics, the clinically exploited nanoparticles in oncology are, e.g., ^99m^Tc-radiolabeled sulfur [1,2] or calcium phytate nanoparticles (to track functional liver reserve or sentinel lymph nodes after solid tumor resection) [3,4], for therapy, e.g., paclitaxel-loaded albumin nanoparticles are used as anticancer agents (Abraxane^®^) [5,6].

Nanoparticles may benefit from the enhanced permeability and retention (EPR) effect of passive accumulation in solid tumor tissue [7,8]. This effect is given by the poor quality of tumor neovasculature with fenestration allowing extravasation of NPs up to ca 200 nm size into solid tumor tissue together with missing lymphatic drainage, also typical for solid tumors, decreasing removal of NPs from such tissue. In addition to this, the EPR effect passive accumulation in solid tumors may be combined with other types of targeting, e.g., with ligands for tissue-specific receptors [9]. Interestingly, it has been shown in preclinical studies that the immunostatus affects the EPR effect, with nanoparticle accumulation being lower in mice lacking a proper immune system [10]. This may be due to altered macrophage density and activity in immunocompromised animals, but the exact reasons for this are not known [11].

For any solid tumor targeting, the nanoparticles must circulate long enough in the bloodstream to reach the tumor. For this, they must be colloidally stable (this is also needed to prevent embolization) and must not be too much scavenged into reticuloendothelial system. Coating NPs with hydrophilic water-soluble polymers can provide both these features to the NPs. Examples of such polymers are poly(ethylene oxide) [12,13,14], poly(2-ethyl-2-oxazoline) [15,16] and poly[*N*-(2-hydroxypropyl)methacrylamide] [17].

Poly(ethylene oxide) is still the “gold standard” for making stealth colloidally stable NPs with a number of clinical applications [12]. However, more recently there is increasing number of reports that this polymer is immunogenic. Antibodies against poly(ethylene oxide) are present in tens of percent of Western population most likely due to overuse of poly(ethylene oxide)-based detergents in everyday products such as liquid soaps, cleaning lotions and shampoos. This may compromise the use of this polymer for such patients. This effect is often referred to as accelerated blood clearance (ABC) phenomenon [18,19]. This polymer is typically synthesized by ring-opening anionic polymerization and only chain ends may be functionalized.

Hydrophilic poly(2-oxazoline)s such as poly(2-ethyl-2-oxazoline) [20] are modern emerging alternatives to poly(ethylene oxide). They are not used in typical everyday products and there are only very little reports about their potential immunogenicity. These polymers are typically synthesized by ring-opening cationic polymerization [21] and the whole polymer chain as well as the chain ends may be functionalized.

Poly[*N*-(2-hydroxypropyl)methacrylamide] is, except for biocompatibilization of nanoparticles, often used for the construction of water-soluble drug delivery systems [17,22]. There are no reports showing its potential immunogenicity. This polymer is typically synthesized by reversible addition−fragmentation chain-transfer polymerization (RAFT) controlled radical polymerization and the whole polymer chain as well as the chain ends may be functionalized.

Hydroxyapatite (HAP) NPs are known to be biodegradable and biocompatible as HAP is the mineral component of bones. Geminal hydroxyalkylidene diphosphonates possess excellent affinity to HAP [23] and number of other inorganic nanoparticles [24] due to their structural similarity to diphosphate, which is present in the crystal lattice of HAP. Radiolabeled hydroxyalkylidene diphosphonates therefore accumulate in bones and are used as bone-seeking radiodiagnostics for bone metastases, fractures and sites of bone remodeling in general [25].

Majority of studies however employ just one type of coating polymer while head-to-head comparison of different biocompatible polymers is largely missing for most biomedical applications related to various diseases. Recently, we published a comparison of antitumor efficacy of poly[*N*-(2-hydroxypropyl)methacrylamide]-based and poly(2-ethyl-2-oxazoline)-based doxorubicin delivery systems in in vivo animal tumor models [26]. We also reported that HAP NPs can be coated with hydrophilic biocompatible polymers using terminal hydroxybisphosphonate groups and efficiently radiolabeled even in situ and in vivo in healthy mice with commercially available bone-seeking radiopharmaceuticals such as ^99m^Tc-hydroxyethylidene diphosphonate (^99m^Tc-HEDP) [23]. The main purpose of this study is to head-to-head compare biological behavior (blood circulation time, biodistribution including solid tumor and organ accumulation and biodegradation) of three leading polymers used as coating materials for biocompatibilization of NPs. We selected HAP NPs as model biodegradable inorganic core-based widely studied NPs. As coating polymers, poly(ethylene oxide), poly(2-ethyl-2-oxazoline) and poly[*N*-(2-hydroxypropyl)methacrylamide] with weight-average molecular weights (*M_w_*) typically used for coating NPs were selected. The polymer-coated radiolabeled NPs were compared in an in vivo animal solid tumor model. To the best of our knowledge, no such study was published in literature in an in vivo animal solid tumor model. This study therefore offers general comparison of in vivo biological applicability of the three most often used biocompatible polymers for coating of inorganic core-based nanoparticles for diagnostics, therapy and theranostics of various diseases such as cancer demonstrating it on in vivo animal model.

## 2. Methods

### 2.1. Synthesis and Characterization of HAP NPs

The HAP NPs were synthesized and coated as reported previously [23]. Briefly, a 0.8 M aqueous solution of (NH_4_)_2_HPO_4_ (98%, Sigma Aldrich, St. Louis, MO, USA) with pH adjusted to 9-11 by the addition of aqueous ammonia was mixed with 1.2 M aqueous solution of Ca(NO_3_)_2_×2H_2_O (98%, Sigma Aldrich, St. Louis, MO, USA) and stirred at ambient temperature for 1 h. Then the NPs were washed twice by repeated centrifugation and dried. Naked HAP NPs were characterized with X-ray diffraction (XRD) (an HZG/4A powder diffractometer; Seifert GmbH, Freiberg i. Sa., Germany), transmission electron microscopy (TEM) (Tecnai Spirit G2 transmission electron microscope; FEI Brno, Czech Republic) and nitrogen adsorption (Gemini VII 2390 system; Micromeritics Instruments Corp. Norcross, GA, USA). The following polymers with terminal hydroxybisphosphonate anchoring groups were used for coating of NPs: poly(ethylene oxide) (*M_w_* 2 kDa - PEG2000 or 5 kDa - PEG5000), poly(2-ethyl-2-oxazoline) (*M_w_* 5 kDa - POX5000 or M_w_ 10 kDa - POX10000) and poly[N-(2-hydroxypropyl)methacrylamide] (*M_w_* 5 kDa - PHPMA5000). For the coating to the polymer solution (0.2 g in 4.4 mL of water) was added 0.07 g of the HAP NPs powder, the mixture was sonicated for 5 min and stirred under ambient conditions for 24 h. Coated NPs were purified by centrifugation and characterized with dynamic light scattering (DLS) (Nano-ZS Zetasizer ZEN3600 Model; Malvern Instruments, Malvern, UK) and thermogravimetric analysis (TGA) (TGA 7 Thermogravimetric Analyzer; Perkin Elmer, Waltham, MA, USA).

### 2.2. Radiolabeling of HAP NPs

The NPs (3 mg) dispersed in water were radiolabeled with ^99m^Tc-HEDP (~100 µL corresponding to 100–150 MBq) as described previously [23]. Radiochemical purity of the radiolabeled HAP NPs was determined by instant thin-layer chromatography on silica gel-impregnated glass fiber sheets (ITLC-SG) using 1 M aqueous ammonium acetate as a mobile phase (Varian, Lake Forest, CA, USA). The retention factor (*R_f_*) of the radiolabeled HAP NPs was zero, while the R_f_ of ^99m^Tc-HEDP was 0.2–0.8.

### 2.3. Cell Cultivation

Human colorectal adenocarcinoma cell line HT-29 (ATCC, Manassas, VA, USA) and human colorectal carcinoma cell line HCT116 (ATCC, Manassas, VA, USA) were cultured in McCoy’s 5A modified medium (Sigma Aldrich, St. Louis, MO, USA) supplemented with 10% fetal bovine serum (Sigma Aldrich, St. Louis, MO, USA) at 37 °C in a 5% carbon dioxide humidified incubator. The cells were subcultured and used for xenografting at a confluency of 70–90%.

### 2.4. Animal Experiments

Animal experiments were conducted in accordance with the regulations and guidelines of the Czech Animal Protection Act (No. 246/1992) and with the approvals of the Czech Ministry of Education, Youth and Sports (MSMT-18724/2016-2) and the Institutional Animal Welfare Committee of the Faculty of Medicine and Dentistry of Palacky University in Olomouc. The studies were performed using female 8–10-week-old Balb/c or SCID mice (Envigo, Horst, The Netherlands). The number of animals was reduced as much as possible (*n* = 3 per group and time point) for all in vivo experiments. The tracer injection and small animal imaging were carried out under 2% isoflurane anesthesia (FORANE, Abbott Laboratories, Abbott Park, IL, USA) to minimize animal suffering and to prevent animal motion.

### 2.5. Ex Vivo Biodistribution

For normal biodistribution experiments, 100 µL of ^99m^Tc-HEDP labeled HAP NPs (~150 µg NPs; 2 MBq/mouse) were retroorbitally (r.o.) injected into the Balb/c mice. The animals were sacrificed by cervical dislocation in selected time points (1, 6 and 24 h) post injection (p.i.). Organs (spleen, pancreas, stomach, intestine, kidneys, liver, heart and lung), blood and muscle and bone tissue were removed and weighted. Activity of the samples was measured in the gamma counter using the respective gamma-energy window and the results expressed as percentage of injected dose per gram tissue (% ID/g).

For biodistribution studies in tumor xenograft bearing mice, female SCID mice were used. For the induction of tumor xenografts, mice were injected subcutaneously near the front shoulder with one million cells (HT-29) in 100 µL McCoy’s 5A modified medium. The tumor growth was continuously monitored by caliperation. When the tumor volume reached ~0.5 cm^3^ (i.e., 2–3 weeks after inoculation of cells), the mice were used for ex vivo biodistribution study. At the day of the experiment, 2 MBq (~150 µg NPs per animal) was administered retroorbitally. At time points 1 and 24 h the mice were sacrificed; the accumulated radioactivity was determined in organs and tumor tissue and the results expressed as percentage of injected dose per gram tissue (% ID/g).

### 2.6. Imaging Studies

The single photon emission computed tomography/computed tomography (SPECT/CT) scans were performed with a small animal PET/SPECT/CT imaging system (Albira, Bruker Biospin Corporation, Woodbridge, CT, USA). During the scans, the mice were anesthetized with 2% isoflurane. Static whole-body SPECT scans of 30 min duration were performed at 1, 3, 6 and 24 h after retroorbital injection of ^99m^Tc-HEDP-labeled HAP NPs (~1.3 mg NPs; 40–60 MBq/mouse). The SPECT scans were followed by a double CT of 20 min (axial FOV 2 × 65 mm, 45 kVp, 400μA, at 400 projections). Reconstruction of the acquired data was performed with the Albira software (Bruker Biospin Corporation, Woodbridge, CT, USA) using the ordered subset expectation maximization and filtered backprojection (FBP) algorithms. All images were prepared using PMOD software, v. 3.307 (PMOD Technologies Ltd., Zurich, Switzerland).

### 2.7. Statistical Analysis

One-way ANOVA followed by Tukey’s post hoc test was used to assess statistical significance. Differences of *p* < 0.05 were considered statistically significant.

## 3. Results

### 3.1. Synthesis and Characterization of HAP NPs

The XRD analysis of synthesized NPs confirmed hexagonal hydroxyapatite structure (PDF 72-1243) with crystallite size 22 to 74 nm, calculated for main crystalline planes {002}, {211}, {300} and {202} (Figure 1A). TEM study of naked HAP NPs revealed elongated flakes with average dimensions 15 nm × 60 nm, in good agreement with XRD (Figure 1B). Naked HAP NPs precipitated in water and PBS within seconds, but coated with polymers, acquired excellent coloidal stability, essential for in vivo applications. The number average diameter of coated HAP NPs was in range 51–82 nm as was measured by DLS (Figure 1C), which was close to the sizes observed by TEM. Therefore, we assume HAP NPs exist in dispersion media as single particles surrounded with polymer corona. BET surface of HAP NPs was 242.2 m^2^/g. The weight loss on thermograms (Figure 1D) was attributed to the decomposition of polymer layer, and the amounts of adsorbed polymer varied in range 5.3 × 10^−8^ mol/m^2^ for PHPMA5000 (the loosest) to 1.0 × 10^−6^ mol/m^2^ for PEG2000 (the densest) [23].

### 3.2. Radiolabeling of HAP NPs

The quality control of radiolabeled polymer-coated HAP NPs was performed after 60 min incubation of ^99m^Tc-HEDP with respective polymer-coated HAP nanoparticles. The radiochemical purity values ranged between 97.5% and 99.9% for all five types of HAP nanoparticles labeled with ^99m^Tc-HEDP (see Figure 2).

### 3.3. Ex Vivo Biodistribution Studies

In healthy Balb/c mice (Figure 3), ^99m^Tc-HEDP labeled NPs showed relatively rapid accumulation in spleen and liver and low blood levels, even at 1h post injection (0.50–1.49% ID/g). The amounts of radioactivity accumulation and retention in spleen and liver slightly differed among tested NPs in time (Figure 4). However, both organs showed highest levels of radioactivity for all NPs in the studied time points (1, 6 and 24 h p.i.). The ex vivo biodistribution profiles of all tested NPs were significantly different to the biodistribution of ^99m^Tc-HEDP. ^99m^Tc-HEDP displayed fast accumulation and retention mainly in bones.

### 3.4. In Vivo Imaging Studies

SPECT/CT imaging of ^99m^Tc-HEDP labeled HAP NPs in normal Balb/c mice (Figure 7) confirmed the results obtained from ex vivo biodistribution studies. All tested NPs showed relatively rapid accumulation and retention of radioactivity mainly in liver and spleen up to 24 h p.i. In the early time points, radioactive signal could be observed also in urinary bladder and heart, especially for PEG coated NPs.

The SPECT/CT imaging of ^99m^Tc-HEDP labeled HAP NPs in tumor xenografted models revealed similar results for all tested NPs. Figure 8 shows SPECT/CT images of ^99m^Tc-PEG5000-HAP in HT-29 tumor-bearing mouse and Figure 9 in HCT116 tumor-bearing mouse. In both tumor models, which were previously confirmed on presence of EPR effect-base solid tumor accumulation [27,28] the ^99m^Tc-HEDP labeled HAP NPs coated with PEG5000 did not visualize the tumor on SPECT in any of the studied time points. The radioactive signal was registered only in heart, liver, spleen and urinary bladder. SPECT/CT images of ^99m^Tc-PEG5000-HAP are presented as an example based on the best tumor-to-blood ratios of this compound from ex vivo biodistribution studies.

## 4. Discussion

In recent years, different organic and inorganic nanoparticles (NPs) such as dendrimers, micelles, liposomes, proteins, polymers, viral capsids, metal oxides (e.g., iron-oxide), gold, quantum dots, zeolites, mesoporous silica, rare earth metals and hydroxyapatite have attracted considerable attention for a variety of medical applications including drug delivery, targeted therapy and molecular imaging [29,30,31]. Inorganic NPs have gained significant attention due to their unique material- and size-dependent physicochemical properties, which may not be feasible for more traditional organic NPs. In particular, characteristics such as chemical inertness, good stability and feasibility of functionalization for molecular imaging make inorganic NPs charming for multimodal imaging of malignant tumor [32].

The main aim of the present study was head-to-head comparison of different biocompatible protective polymer coatings for EPR effect-based tumor-targeted nanoparticle radiodiagnostics based on inorganic hydroxyapatite. As the EPR effect is relatively universal for many solid tumors, such radiodiagnostics offer wide potential for oncology. We selected hydroxyapatite as model nontoxic biodegradable nanoparticle material, which might be radiolabeled with bone-seeking clinically approved radiopharmaceuticals. The feasibility of such radiolabeling was proven by our study. The polymers to be compared were selected on the basis of their wide applicability and we utilized our previously developed hydroxybisphosphonate-based anchoring strategy [23]. Except for multimodal imaging, we also performed exact ex vivo radioactivity quantification to follow true biodistribution in healthy and tumor-bearing mice. Naked HAP nanoparticles without polymer were not included in the animal studies as they are not colloidally stable in blood plasma and immediately precipitate causing embolization.

Ex vivo biodistribution studies in healthy animals showed that ^99m^Tc-HEDP accumulated and retained mainly in bones and partly also in the liver. This is in full agreement with previous reports of van Leeuwen et al. [33] and Khmelinskii et al. [34]. In contrast, ^99m^Tc-HEDP labeled HAP NPs accumulated rapidly in spleen, liver and bones and were retained in these organs until 24 h after injection, probably due to certain extent of entrapment into reticuloendothelial system, for which is this biodistribution profile typical. The difference between biodistribution of free and HAP-NPs bound ^99m^Tc-HEDP also shows that ^99m^Tc-HEDP remains bound to HAP even during circulation in bloodstream in vivo. All the used coating polymers provide colloidal stability to the nanoparticles (naked HAP NPs without polymer coating cannot be injected into bloodstream due to aggregation and subsequent embolization). In tumor-bearing animals, we have shown that polymer coating have a crucial effect on the in vivo fate and tumor targeting of the nanoparticles. Whereas NPs coated with poly(ethylene oxide) (PEG) and poly(2-ethyl-2-oxazoline) (POX) revealed comparable pharmacokinetics and tumor targeting, NPs coated with poly[*N*-(2-hydroxypropyl)methacrylamide] (PHPMA) displayed slightly higher retention in lungs and almost no uptake in the tumor, probably due to too fast uptake into reticuloendothelial system given by worse protection of the HAP NPs core by PHPMA compared to the other polymers involved not giving the system enough blood circulation time to allow solid tumor accumulation. These results show similar high quality of polymer coating protecting the HAP nanoparticles from unwanted interactions in bloodstream and suitable in vivo stability of the nanoparticles on the nanoparticles. As all polymers possess the same HAP-anchoring terminal hydroxybisphosphonate moiety, the differences among the PHMPA-coated HAP NPs and the POX- and PEG-coated NPs is purely given by the nature of the coating polymer. We did not show the accelerated blood clearance (ABC) phenomenon for the PEG-coated HAP NPs sometimes observed for the PEG-based systems, probably because the mice were not pre-exposed to PEG-based system.

We have also not observed significant dependence of the biological behavior on the molecular weight of the polymer used for coating, so most plausibly event he lower molecular weight polymer is able to provide sufficiently dense coating to prevent unwanted interactions with the biological environment. Among the ^99m^Tc-HEDP labeled HAP NPs tested ^99m^Tc-PEG5000-HAP revealed the most favorable tumor-blood ratio (6.14 for 24 h p.i.) and ^99m^Tc-PHPMA5000-HAP (0.99 for 24 h p.i.) the most unfavorable. Tumor-muscle ratios were in the range of 2.52 – 6.91 for all PEG and POX coated HAP NPs at 24 h post injection. ^99m^Tc-PHPMA5000-HAP showed again the lowest tumor-muscle ratio (1.01 for 24 h p.i.) from tested NPs. Based on these findings ^99m^Tc-HEDP labeled PEG and POX HAP NPs were used for SPECT/CT imaging of tumor-bearing mice and PHPMA coated HAP NPs were excluded from further experiments.

SPECT/CT scans performed with healthy mice injected with ^99m^Tc-HEDP labeled HAP NPs coated with PEG or POX revealed pharmacokinetic properties as expected from ex vivo biodistribution studies: high activity uptake was observed in liver and spleen (most plausibly due to the uptake into reticuloendothelial system) during all imaging time points and certain activity was present in heart and urinary bladder (most likely due to biodegradation of the HAP NPs core releasing free ^99m^Tc-HEDP, which is then eliminated by kidneys into urine, in particular early after the injection. To investigate the possibility of tumor imaging with ^99m^Tc-HEDP labeled HAP NPs, SPECT/CT imaging in mice bearing human colorectal adenocarcinoma (HT-29) or human colorectal carcinoma (HCT116) was performed. Although, ex vivo biodistribution studies in HT-29 xenografted mice displayed clear uptake of ^99m^Tc-PEG5000-HAP in the tumor and highest tumor-blood ratio among studied HAP NPs, we were not able to image either HT-29 or HCT116 tumor in mice with ^99m^Tc-PEG5000-HAP using small animal SPECT/CT imaging system in any of studied time points (1, 3, 6 and 24 h) after injection. The same results were obtained also for other PEG and POX coated HAP NPs.

Whereas the SPECT imaging with ^99m^Tc labeled HAP NPs under study was not successful to visualize tumors, we started to reflect on the use of positron emitters for the radiolabeling of HAP NPs for positron emission tomography (PET) imaging. The main advantage of PET over SPECT is its spatial resolution, the ability of absolute quantification and higher sensitivity [35], which could give a better chance to image the tumor tissue using studied HAP NPs. It is well known that several positron-emitting radionuclides (e.g., fluorine-18 and zirconium-89) are considered as bone-seekers due to their high affinity to hydroxyapatite. Zheng et al. [36] have investigated HAP nanoparticles labeled with ^18^F in vivo and determined pharmacokinetics and biodistribution of ^18^F-HAP NPs in rabbits up to 160 min post injection showing relatively long circulation period of HAP NPs. Although our ^99m^Tc labeled HAP NPs did not show extremely high retention in blood (maximum 1.49 ± 0.5% ID/g for ^99m^Tc-PEG5000-HAP in healthy mice 1 h p.i.) short half-life of ^18^F (*t*_1/2_ = 109.7 min) could be a limiting factor for the use of this radionuclide and longer-lived zirconium-89 (*t*_1/2_ = 3.3 d) will probably be a better choice for labeling of HAP NPs. Post-labeling in vivo approach with bone-seeking radiopharmaceuticals [23] may also help in tumor visualization by means of nuclear imaging techniques.

## 5. Conclusions

We have shown that HAP NPs coated with different polymers of varying molecular weights can be radiolabeled with ^99m^Tc-HEDP. Poly(2-ethyl-2-oxazoline) and poly(ethylene oxide) coated HAP NPs revealed comparably similar pharmacokinetics enabling solid tumor targeting in mice, while poly[*N*-(2-hydroxypropyl)methacrylamide] coated HAP NPs displayed significantly lower uptake in tumor tissue. Unfortunately, SPECT imaging was not successful in visualizing the tumor. We did not observe strong effect of molecular weight of the coating polymers on tumor and organ accumulation, blood circulation time, biodistribution and biodegradation of HAP NPs under the study.

## Figures and Tables

**Figure 1 nanomaterials-10-01690-f001:**
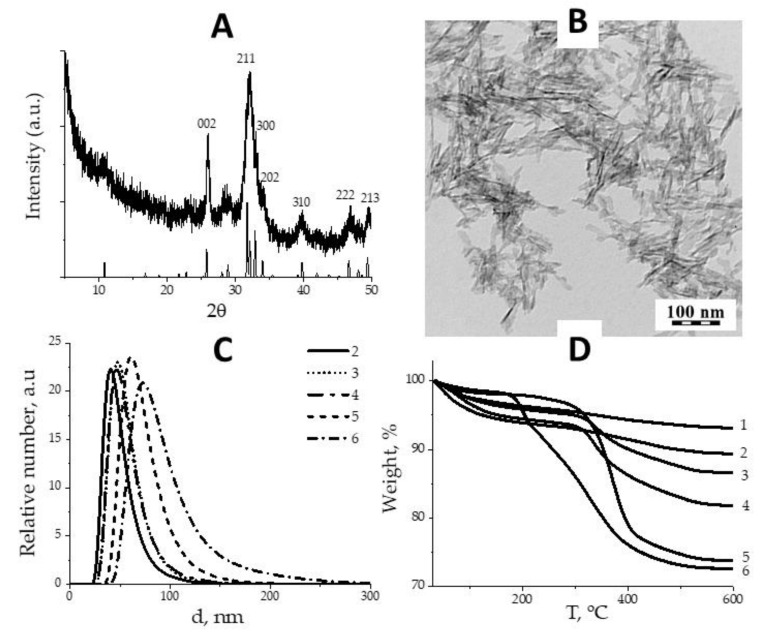
XRD pattern of hydroxyapatite (HAP) nanoparticles (NPs) (**A**); TEM micrograph of HAP NPs (**B**); number weighted particles size distributions (**C**) and thermograms (**D**) of HAP NPs coated with: (1) naked, (2) PHPMA5000, (3) POX5000, (4) POX10000, (5) PEG5000, (6) PEG2000.

**Figure 2 nanomaterials-10-01690-f002:**
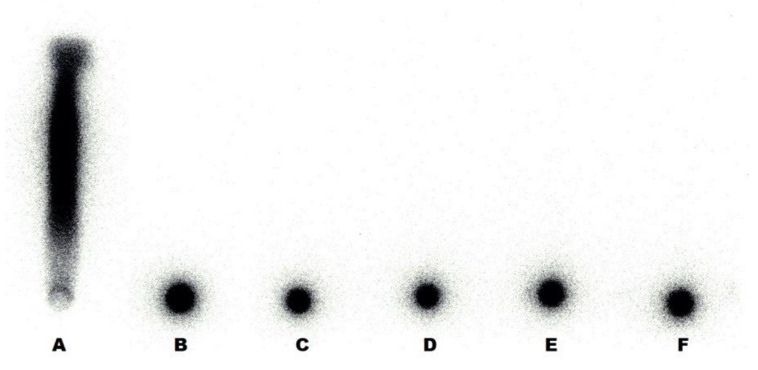
Instant thin-layer chromatography on silica gel impregnated glass fibres (ITLC-SG) radiochromatogram of ^99m^Tc-hydroxyethylidene diphosphonate (^99m^Tc-HEDP) and nanoparticles labeled with ^99m^Tc-HEDP. ^99m^Tc-HEDP (**A**), ^99m^Tc-PEG2000-HAP (**B**), ^99m^Tc-PEG5000-HAP (**C**), ^99m^Tc-POX5000-HAP (**D**), ^99m^Tc-POX10000-HAP (**E**) and ^99m^Tc-PHPMA5000-HAP (**F**).

**Figure 3 nanomaterials-10-01690-f003:**
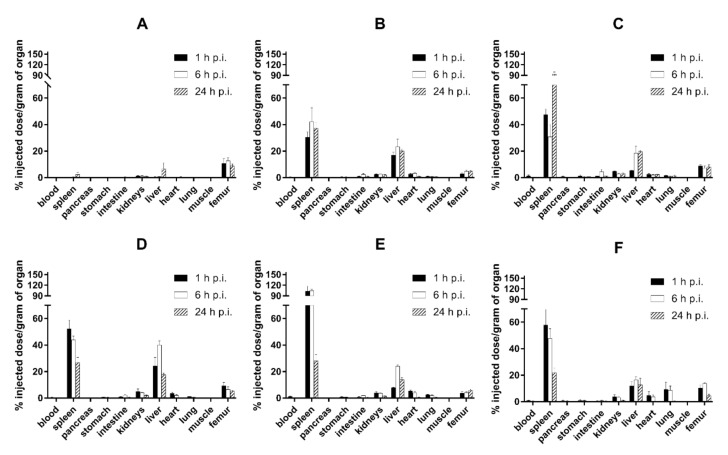
Biodistribution of r.o. injected ^99m^Tc-HEDP (**A**), ^99m^Tc-PEG2000-HAP (**B**), ^99m^Tc-PEG5000-HAP (**C**), ^99m^Tc-POX5000-HAP (**D**), ^99m^Tc-POX10000-HAP (**E**) and ^99m^Tc-PHPMA5000-HAP (**F**) in healthy Balb/c mice 1, 6 and 24h post injection (p.i.) (*n* = 3 animals per time point).

**Figure 4 nanomaterials-10-01690-f004:**
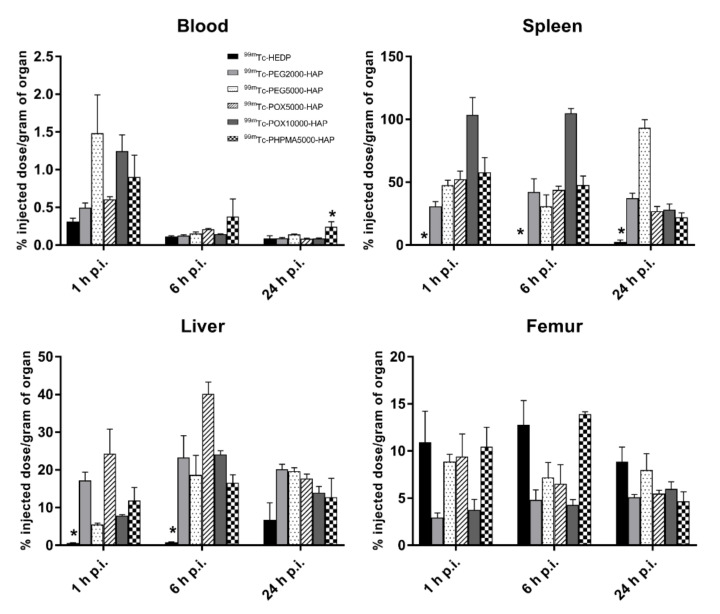
Uptake of ^99m^Tc-HEDP, ^99m^Tc-PEG2000-HAP, ^99m^Tc-PEG5000-HAP, ^99m^Tc-POX5000-HAP, ^99m^Tc-POX10000-HAP and ^99m^Tc-PHPMA5000-HAP NPs in blood, spleen, liver and femur of healthy Balb/c mice 1, 6 and 24 h p.i. (*n* = 3 animals per time point); * indicates statistical difference (*p* < 0.05). Biodistribution of ^99m^Tc-HEDP labeled HAP NPs in tumor xenograft model (Figure 5) revealed certain uptake in the HT-29 tumor for NPs coated with PEG and POX, ^99m^Tc-HEDP labeled HAP NPs coated with PHPMA did not show any specific uptake in tumor tissue and were excluded from further studies (Figure 6). ^99m^Tc-HEDP labeled HAP NPs coated with PEG5000, POX5000 and POX10000 displayed similar tumor-blood ratios (Table 1) at 24h p.i. with the highest value (6.14) for NPs coated with PEG5000. All tested NPs showed also satisfying tumor-background (muscle) ratios (1.49–6.91) for both time points, except for NPs coated with PHPMA (1.30, 1.01). Slight differences among studied NPs were observed in the retention in blood, especially in the short time (1 h) after injection.

**Figure 5 nanomaterials-10-01690-f005:**
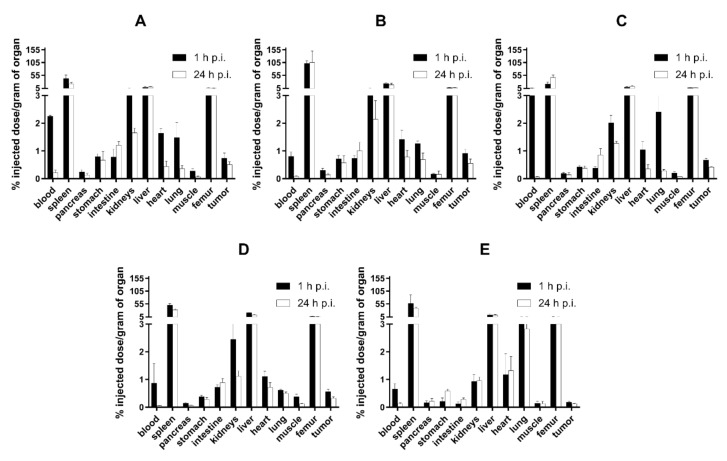
Biodistribution of retroorbitally (r.o.) injected ^99m^Tc-PEG2000-HAP (**A**), ^99m^Tc-PEG5000-HAP (**B**), ^99m^Tc-POX5000-HAP (**C**), ^99m^Tc-POX10000-HAP (**D**) and ^99m^Tc-PHPMA5000-HAP (**E**) in tumor (HT-29) SCID mice 1 and 24 h p.i. (*n* = 3 animals per time point).

**Figure 6 nanomaterials-10-01690-f006:**
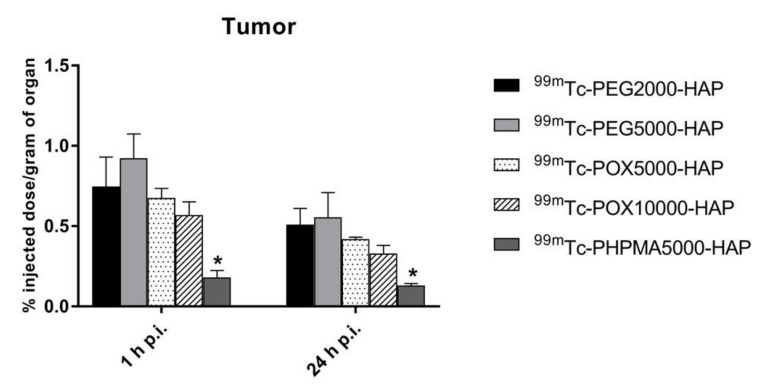
Uptake of ^99m^Tc-PEG2000-HAP, ^99m^Tc-PEG5000-HAP, ^99m^Tc-POX5000-HAP, ^99m^Tc-POX10000-HAP and ^99m^Tc-PHPMA5000-HAP NPs in HT-29 tumors in SCID mice 1 and 24 h p.i. (*n* = 3 animals per time point). * indicates statistical difference (*p* < 0.05).

**Figure 7 nanomaterials-10-01690-f007:**
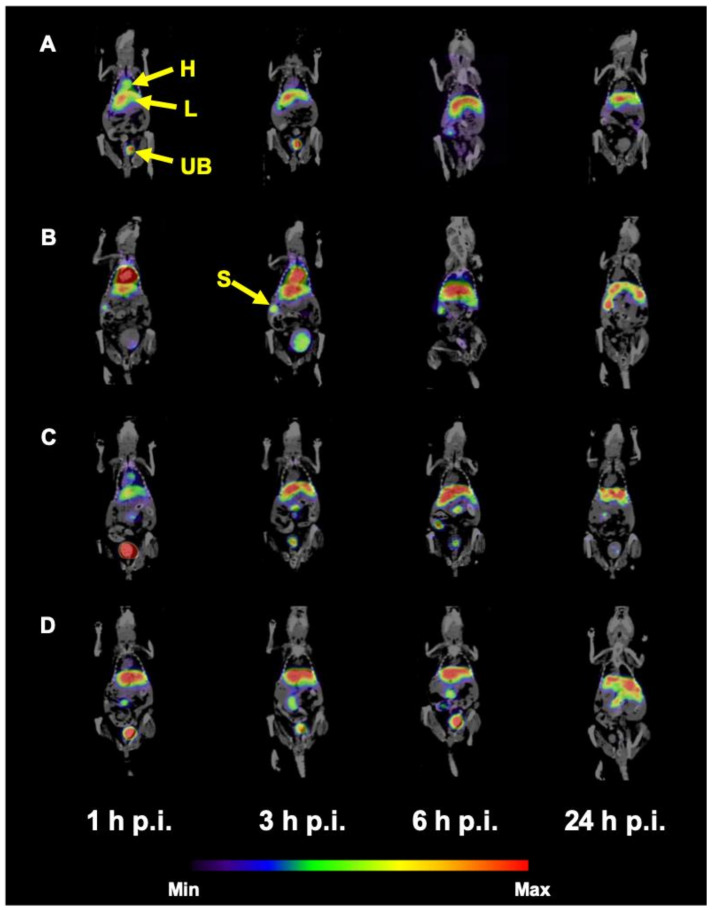
Coronal slices of SPECT/CT imaging of ^99m^Tc-HEDP labeled HAP NPs in Balb/c mice 1, 3, 6 and 24h p.i. after r.o. injection (^99m^Tc-PEG2000-HAP (**A**), ^99m^Tc-PEG5000-HAP (**B**), ^99m^Tc-POX5000-HAP (**C**), ^99m^Tc-POX10000-HAP (**D**); H = heart, L = liver, UB = urinary bladder, S = spleen).

**Figure 8 nanomaterials-10-01690-f008:**
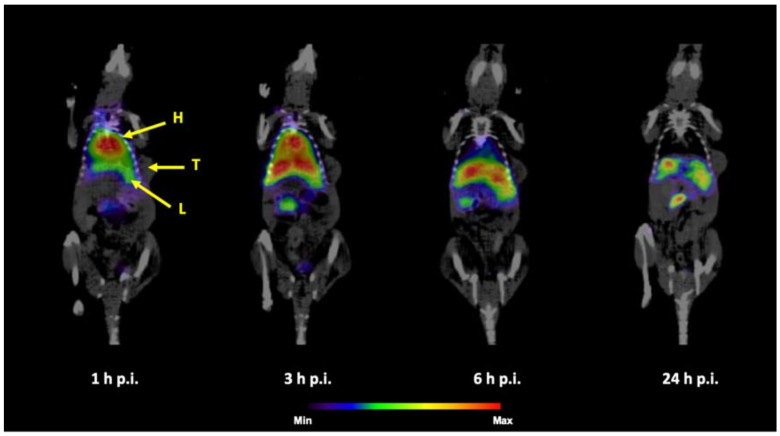
Coronal slices of SPECT/CT imaging of ^99m^Tc-PEG5000-HAP in HT-29 tumor-bearing mouse 1, 3, 6 and 24h p.i. after r.o. injection (H = heart, L = liver, T = tumor).

**Figure 9 nanomaterials-10-01690-f009:**
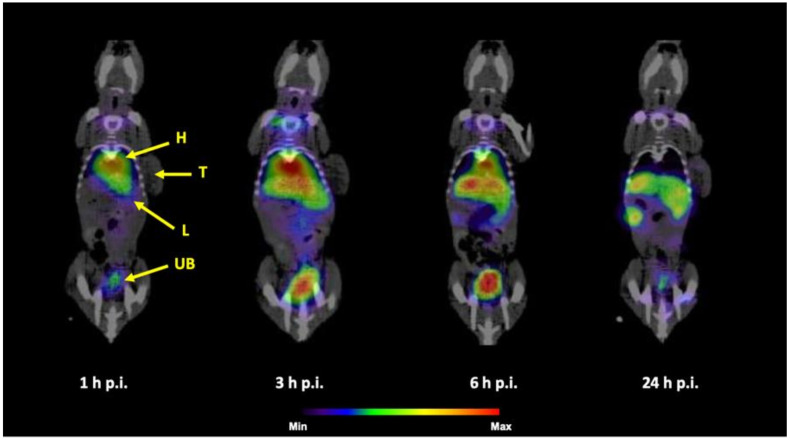
Coronal slices of SPECT/CT imaging of ^99m^Tc-PEG5000-HAP in HCT116 tumor-bearing mouse 1, 3, 6 and 24h p.i. after r.o. injection (H = heart, L = liver, T = tumor, UB = urinary bladder).

**Table 1 nanomaterials-10-01690-t001:** Tumor-background and tumor-excretory organs ratios obtained in HT-29 tumor-bearing mice.

Ratio	^99m^Tc-PEG2000-HAP		^99m^Tc-PEG5000-HAP		^99m^Tc-POX5000-HAP		^99m^Tc-POX10000-HAP		^99m^Tc-PHPMA5000-HAP	
	1 h p.i.	24 h p.i.	1 h p.i.	24 h p.i.	1 h p.i.	24 h p.i.	1 h p.i.	24 h p.i.	1 h p.i.	24 h p.i.
**tumor/blood**	0.33	2.38	1.15	6.14	0.16	5.73	0.66	5.52	0.27	0.99
**tumor/muscle**	2.69	6.91	5.23	3.46	3.28	5.14	1.49	2.52	1.30	1.01
**tumor/kidney**	0.25	0.31	0.27	0.26	0.33	0.33	0.23	0.29	0.19	0.14
**tumor/liver**	0.09	0.05	0.04	0.03	0.08	0.03	0.03	0.03	0.02	0.13

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
