# Peer review of "Head-To-Head Comparison of Biological Behavior of Biocompatible Polymers Poly(Ethylene Oxide), Poly(2-Ethyl-2-Oxazoline) and Poly[N-(2-Hydroxypropyl)Methacrylamide] as Coating Materials for Hydroxyapatite Nanoparticles in Animal Solid Tumor Model"

_nanomaterials, 2020, doi:10.3390/nano10091690_

Round 1

Reviewer 1 Report

Dear authors even I appreciate your work I do not think you can discuss about such Nps behaviour without providing their characterisation. My opinion is that Nanomaterials cannot publish a paper that completely lacks in any morphologic and structural characterisation of the Nps you used.  My suggestion is that you integrate your paper with at least two paragraphs concerning morphological and characterisation methods, results and discussions.  

Author Response

The requested morphologic and structural characterisation of the NPs was added to the manuscript in „track changes“ mode.

Reviewer 2 Report

The main purpose of this study is to compare biological behavior of three polymers commonly used as “stealth” coating materials for biocompatibilization of NPs  in in vivo animal solid tumor model. Radiolabeled biodegradable hydroxyapatite was used as NPs core. The polymers were anchored to the NPs core by hydroxybisphosphonate end groups. The results showed that for solid tumor accumulation it is better for poly(2-ethyl-2-oxazoline) and poly(ethylene oxide) in the ex vivo study, but not for poly[N-(2-hydroxypropyl)methacrylamide].

If these polymers are to be used for drug delivery, there is a quite a concern that accumulation of NPs in tumor was so low compared with other normal organs in the SPECT studies. Although PET may be an alternative imaging method to improve the detection sensitivity (maybe), still, the fundamental issue of tumor uptake vs normal organs accumulation still needs to be addressed. It may be helpful to address this issue with targeted drug delivery model, i.e., these 3 polymer coated NPs linked with ligands targeting specific biomarkers in the tumors. With targeted NPs delivery in tumor bearing animal model, there will be higher NPs accumulation in the tumors. Such a comparison of NPs with various polymer coating compositions accumulation in tumor will be more useful and meaningful.

Author Response

The main purpose of this study is to head-to-head compare biological behavior (blood circulation time, biodistribution including solid tumor and organ accumulation and biodegradation) of three leading polymers used as coating materials for biocompatibilization of NPs. This study therefore offers general comparison of applicability of three most often used biocompatible polymers for coating of inorganic core-based nanoparticles. The aim of this study is not focused on tumor targeting itself (e.g., ligand-based), but rather on biocompatibility and general behavior of the NPs in the tumor-bearing  organism in the scope of head-to-head comparison of the polymers in the organism and select the most suitable coating polymer. Attachment of tumor-targeting ligand would just complicate interpretation of the data and is not the aim of the study. To make this fully clear and avoid possible misunderstanding, we have reformulated the end of the Introduction section accordingly in „track changes“ mode.

The sufficiently long-lived SPECT radionuclide 99mTc was used to enable the same way of radiolabeling for precise ex vivo quantification and in vivo imaging. We agree that employing PET could improve the sensitivity, however in this study with the main focus on the comparison of biological behavior of different coating polymers it would just complicate radiolabeling procedures and would bring no additional value.

Round 2

Reviewer 1 Report

no further suggestion

Reviewer 2 Report

Good revision.